# Epigenetic landscape of the H3K27me3 mark in macrophages transformed by *Theileria annulata*
Takaya Sakura [1,2,3,4,9], Shahin Tajeri [1,8,9], Zineb Rchiad[1,5], Hifzur R. Ansari [5], Abhinav Kaushik[5], Tobias Mourier[5], Arnab Pain [5,6], Michel Wassef [7] ✉ & Gordon Langsley [1,2] ✉

*Theileria annulata* is an obligate intracellular parasite that induces bovine leukocyte transformation leading to uncontrolled proliferation and heightened dissemination of infected leukocytes. Early passage *T. annulata*-transformed macrophages are virulent, but with long-term culture, they become attenuated for dissemination. To investigate this phenotype from an epigenetic perspective, we focused on the important repressive histone marks, tri-methylated lysine 27 of histone H3 (H3K27me3) catalyzed by the Polycomb Repressive Complex 2. ChIP-seq revealed that the genomic distribution of H3K27me3 is heavily remodeled in attenuated macrophages, with many genes transitioning from a focal H3K27me3 peak around the transcription start site to larger chromatin domains. RNA-seq analysis following PRC2 inhibitor treatment reveals that fewer genes are derepressed in attenuated macrophages than in virulent macrophages, suggesting that broader H3K27me3 profiles do not systematically translate into increased gene silencing activity. Our findings shed light on the mechanisms underlying the dysregulation of epigenetic modifications in *Theileria*-induced leukocyte transformation.

Apicomplexan parasites *Theileria annulata* and *T. parva* are transmitted by ticks and infect bovine leukocytes, causing tropical theileriosis primarily in Asian countries and East Coast Fever (ECF) in Africa[1,2]. Macroschizonts differentiated from tick-injected sporozoites induce tumorigenesis of infected leukocytes, and dissemination of transformed leukocytes is mainly responsible for the pathology of these livestock diseases. Despite the huge economic impact of this disease in the veterinary field, available vaccines suffer from several drawbacks, such as the requirement for quality control and cold chain[3,4]. In addition, parasites resistant to the theileriacidal drug buparvaquone are emerging in endemic areas[5,6]. New vaccines and drugs are therefore needed to control these widespread livestock diseases and to reduce their economic burden in endemic areas.

A specific feature of *T. annulata* and *T. parva* infections is the transformation of their host leukocytes that allows the parasite population to grow due to uncontrolled host cell proliferation[7,8]. The mechanisms underpinning *Theileria annulata*-mediated transformation—i.e., how parasites modulate host gene expression and hijack their proliferation—rely on host–parasite interactions. For instance, the IκB kinase (IKK) complex is recruited to the surface of the parasite, resulting in nuclear factor kappa B (NF-κB) being constitutively activated to induce expression of anti-apoptotic genes[9]. In addition, *Theileria* infection inhibits nuclear translocation of p53, a well-known tumor suppressor, and blocks p53-driven expression of pro-apoptotic genes[10,11]. Meanwhile, c-Jun N-terminal Kinase (JNK2) interacts with a GPI-anchored macroschizont surface protein (p104), while JNK1 translocates to the nucleus to activate c-Jun to drive transcription of the matrix metalloproteinase 9 (*mmp*9) gene and enhance tumor dissemination[12–14]. Parasite secreted peptidyl-prolyl isomerase PIN1 also contributes to c-Jun stabilization that maintains the proliferative phenotype of *Theileria*-infected leukocytes[15]. Subversion of host cell signal transduction pathways augments *c-myc*-driven transcription of anti-

[1]Biologie Comparative des Apicomplexes, Institut Cochin, Paris, France. [2]Université de Paris-Cité, INSERM U1016, CNRS UMR 8104, Paris, France. [3]Department of Molecular Infection Dynamics, Shionogi Global Infectious Diseases Division, Institute of Tropical Medicine (NEKKEN), Nagasaki University, Nagasaki, Japan. [4]School of Tropical Medicine and Global Health, Nagasaki University, Nagasaki, Japan. [5]Pathogen Genomics Laboratory, Biological and Environmental Sciences and Engineering (BESE) Division, King Abdullah University of Science and Technology (KAUST), Thuwal, Saudi Arabia. [6]International Institute for Zoonosis Control (IIZC), Institute for Vaccine Research and Development (IVReD), Hokkaido University, Sapporo, Japan. [7]Institut Curie, INSERM U934/CNRS UMR 3215, Paris Sciences et Lettres Research University, Sorbonne Université, Paris, France. [8]Present address: Institute for Parasitology and Tropical Veterinary Medicine, Freie Universität Berlin, Berlin, Germany. [9]These authors contributed equally: Takaya Sakura, Shahin Tajeri. ✉e-mail: michel.wassef@curie.fr; gordon.langsley@inserm.fr

apoptotic genes[16–18]. *Theileria* transformation therefore shares several similarities with human carcinogenesis, where epigenetic (i.e., not involving changes of the DNA sequence) alterations contribute to cancer development. For example, *Theileria*-induced tumor dissemination is regulated by changes in TGF-β2 levels, and dissemination of attenuated macrophages is restored by the addition of exogenous TGF-β2, consistent with a role for epigenetic regulation of cancer-related genes[19–21].

Many types of post-translational modifications on histone tails have been shown to regulate chromatin dynamics and transcription[22]. Changes of the chromatin state have been reported in *Theileria*-infected leukocytes. Tri-methylated lysine 4 of histone H3 (H3K4me3) is a mark correlating with active transcription enriched upstream of the *mmp-9* gene in the *Theileria*-infected B-cell lymphosarcoma cell line (TBL3), and transcriptional activation of *mmp-9* is associated with heightened tumor dissemination[23,24]. Tri-methylated lysine 27 of histone H3 (H3K27me3) is a key chromatin modification catalyzed by the polycomb repressive complex 2 (PRC2)[25]. H3K27me3 is associated with transcriptional silencing, acting in part as a docking site for CBX-containing PRC1 complexes. During embryonic development and throughout adult life, H3K27me3 plays a key role in maintaining transcriptional silencing of genes encoding cell fate regulators[26]. Furthermore, the PRC2 complex is altered in several types of tumors, acting either as a tumor suppressor or an oncogene[27].

*Theileria annulata*-infected macrophages are fully transformed and, as such, can be maintained in culture indefinitely. Early passage *T. annulata*-transformed macrophages are virulent (disseminating), but virulence decreases with long-term (>300) passage, and infected cells become attenuated for dissemination[28,29]. In the present study, using the Polycomb-specific H3K27me3 histone mark as a paradigm, we investigate the evolution of the epigenetic landscape between virulent and attenuated *T. annulata*-infected macrophages. We show that the distribution of H3K27me3 in attenuated macrophages is profoundly altered and that attenuated cells become less responsive to PRC2 inhibition compared to virulent macrophages. However, some tumor suppressor genes still respond to PRC2

inhibitor treatment. Our study, therefore, reveals profound changes in the chromatin landscape that accompany the transition from virulent to attenuated states.

## Results

### *Theileria annulata* parasites within transformed host leukocytes are not stained by an H3K27me3-specific antibody

When comparing the amino acid sequence of the tail of histone H3 between various parasite species, we found that the amino acid sequence surrounding Lys27 is highly conserved between mammals and *Plasmodium falciparum* and *Toxoplasma gondii* (Fig. 1a). Accordingly, H3K27me3 is present in *T. gondii* tachyzoites and has been detected in sexual stage II gametocytes in *P. falciparum*[30–33]. We noticed that there is an S>T change at position 28 of *Theileria* parasite histone H3, raising the question of whether H3K27 is tri-methylated in *Theileria* species. As expected, immunofluorescence analysis using an antibody specific for H3K27me3 stained bovine host nuclei, whereas *Theileria* nuclei were not stained (Fig. 1b). Whether this is due to the absence of the mark and/or a failure of the antibody to recognize it (due to the H3S28>T change) is unclear. By contrast, specific antibodies to H3K4me3 stained both parasite and host cell nuclei. Since H3K27me3 is not detected in the parasite by the antibody we used, the following analyses of H3K27me3 reflect only the bovine histone mark.

### Global levels and distribution of H3K27me3 are altered in attenuated macrophages

The fact that attenuated macrophages have a low dissemination potential led us to hypothesize that oncogenes activated as a consequence of *Theileria* infection might be transcriptionally downregulated through mechanisms promoting a repressed chromatin state. To this end, we quantified H3K27me3 abundance in the nuclear fraction of both virulent and attenuated macrophages (Fig. 2a). The amount of H3K27me3 was estimated to be 1.58-fold higher in attenuated macrophages compared to virulent macrophages, whereas H3K4me3 was 0.66-fold lower in attenuated

**Fig. 1 | Localization of H3K27me3 and H3K4me3 marks in *Theileria*-infected virulent and attenuated macrophages. a** Amino acid sequence alignment of Histone H3 tail among Apicomplexan parasites and *Homo sapiens*. TA07840 sequence is used for *Theileria annulata*. **b** Immunofluorescence staining with H3K27me3 and H3K4me3 (green). The monoclonal p104 antibody labels a parasite surface protein p104 (red) and DAPI-stained nuclei (blue). H3K27me3 antibody only stains host nuclei. Tri-methylation on lysine 4 is conserved between *T. annulata* and bovine host nuclei. Scale bar: 10 μm.

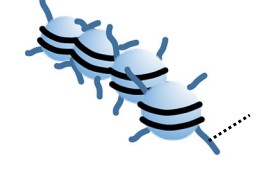

**a**

**Histone H3 tail**

| | 1 4 27 34 |
|---|---|
| *T. annulata* | ART**K**QTARKSTGGKAPRKQLASKAAR**K**TAPVTGG |
| *T. parva* | ART**K**QTARKSTGGKAPRKQLASKAAR**K**TAPVTGG |
| *T. orientalis* | ART**K**QTARKSTGGKAPRKQLASKAAR**K**TAPVTGG |
| *P. falciparum* | ART**K**QTARKSTAGKAPRKQLASKAAR**K**SAPISAG |
| *T. gondii* | ART**K**QTARKSTGGKAPRKQLASKAAR**K**SAPMSGG |
| *H. sapiens* | ART**K**QTARKSTGGKAPRKQLATKAAR**K**SAPATGG |

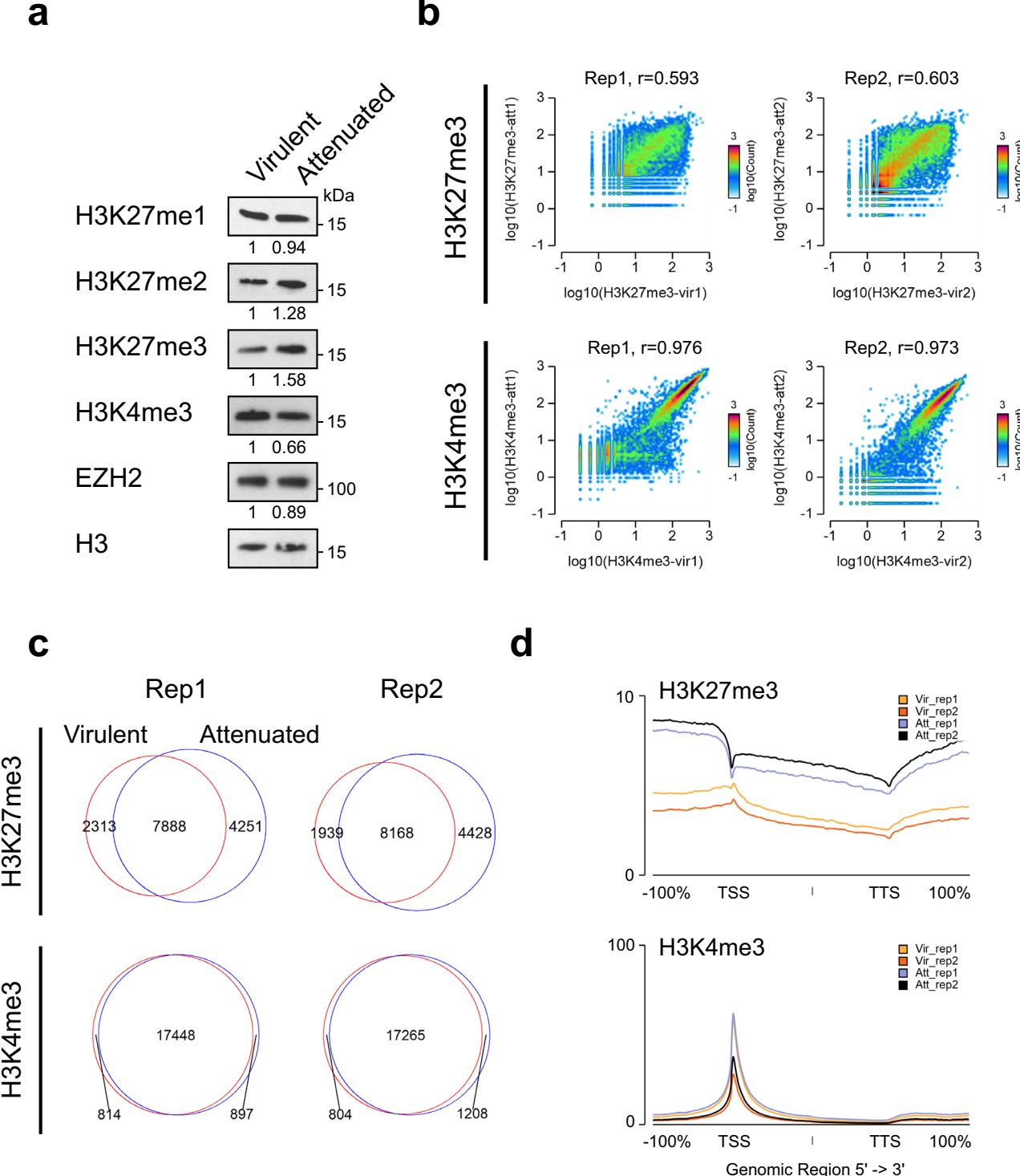

**Fig. 2 | Genome-wide H3K27me3 distribution dramatically changes in attenuated macrophages. a** Comparative abundance of histone marks and EZH2 in virulent and attenuated macrophages. Each band was normalized to H3, used as a loading control. Relative intensity is shown below each band. **b** 2D density plots comparing H3K27me3 and H3K4me3 signal between virulent and attenuated macrophages in each of the two biological replicates. Read counts for each gene in virulent (*X*-axis) and attenuated (on *Y*-axis) macrophages are plotted. The *r* values represent the Pearson correlation coefficients. **c** Venn diagrams showing the overlap between virulent and attenuated macrophages for H3K27me3- and H3K4me3-

positive genes. Peaks of H3K27me3 and H3K4me3 were associated with the *B. taurus* transcripts (UMD3.1.1), and signal intensity around the TSS ± 2 kb was quantified using Easeq (52). **d** Average H3K27me3 and H3K4me3 signal intensities at all annotated *Bos taurus* genes (bosTau8). The *X*-axis represents a window centered on each gene, spanning from the transcription start site (TSS) to the transcription termination site (TTS) with an additional ±100% flanking region. Yellow/orange lines represent signals from virulent macrophages, whereas purple/black lines represent those from attenuated macrophages.

macrophages. The increase in H3K27me3 levels in attenuated macrophages was confirmed by quantifying fluorescence signals from individual cells in confocal microscopy images (Supplementary Fig. 1). Interestingly, the expression of EZH2, the main PRC2 catalytic subunit, appeared unchanged between virulent and attenuated macrophages.

We next investigated the genome-wide distribution of the two histone marks in *Theileria*-infected macrophages by chromatin immunoprecipitation sequencing (ChIP-seq). *Drosophila melanogaster* chromatin was used as a spike-in to normalize the abundance of the histone marks[34]. Consistent with immunoblotting results, the density of H3K27me3 around transcriptional start sites (TSS) was generally higher in attenuated than virulent macrophages, while H3K4me3 was lower (Supplementary Fig. 2a). H3K27me3 was also poorly correlated between virulent and attenuated macrophages at the genome-wide level ($r = 0.593$ and $0.603$ for replicates 1 and 2, respectively), contrasting with the high degree of H3K4me3 correlation ($r = 0.976$ and $0.973$ for replicates 1 and 2 respectively) (Fig. 2b). A substantial fraction of H3K27me3-positive genes were specific to either virulent or attenuated macrophages, while a relatively smaller proportion of H3K4me3-positive genes differed between the two cell states (Fig. 2c). Strikingly, plotting the average H3K27me3 signal revealed that while the mark was increased overall over the gene bodies of attenuated cells, its relative distribution around the TSS was markedly different between virulent and attenuated cells (Fig. 2d). In virulent macrophages, H3K27me3 showed a typical enrichment around the TSS. However, in attenuated macrophages, H3K27me3 enrichment was no longer maximal at the TSS. Instead, the mark accumulated most prominently in intergenic regions (upstream of TSS and downstream of transcription termination sites). This contrasts with H3K4me3, which accumulated sharply around the TSS and showed a similar distribution pattern between virulent and attenuated macrophages.

We further defined 3 clusters of genes according to H3K27me3 enrichment (Fig. 3a): virulent-positive (cluster 1), attenuated-positive (cluster 2), both positive (cluster 3). While virulent-specific H3K27me3 signal showed a sharp enrichment around the TSS, attenuated-specific H3K27me3 signal showed a wide distribution around the TSS, with a maximal enrichment upstream of the TSS. This difference was also obvious for genes showing H3K27me3 enrichment in both conditions (Fig. 3a). Average transcript levels in virulent and attenuated macrophages for each H3K27me3 cluster showed no statistically significant difference as a group, while differential gene expression analysis identified a subset of cluster 2 genes that were downregulated in attenuated versus virulent macrophages (Fig. 3b)[35]. One of the genes highly enriched for H3K27me3 in cluster 2 of attenuated macrophages is the Src Kinase Associated phosphoprotein 2 (*SKAP2*), which is associated with macrophage cell adhesion[36]. It is covered by a large domain of H3K27me3 extending over 500 kb, in contrast to a low level concentrated at the TSS found in virulent macrophages (Fig. 3c). These differences are paralleled by a corresponding downregulation of *SKAP2* expression in attenuated macrophages[35]. Patterns of H3K27me3 and H3K4me3 distribution around representative genes of other clusters, such as TBX5 (Cluster 1) and WNT3 (Cluster 3), further illustrate the broader spreading of H3K27me3 in attenuated cells compared to virulent cells (Fig. 3c). Altogether, these data show that the attenuated phenotype is characterized by a dramatic reconfiguration of H3K27me3 distribution.

### Limited effect of PRC2 inhibition on tumor-related phenotypes
In light of the widespread re-distribution of H3K27me3 in attenuated cells, we next investigated the requirement for the mark in the transformed and attenuated phenotypes. We treated both virulent and attenuated macrophages with a specific PRC2 inhibitor (UNC1999) or its inactive analog (UNC2400)[37,38]. Ten days of treatment with UNC1999 led to an acute loss of H3K27me3 in both cell lines (Fig. 4a). We then performed fibronectin-binding assays on the PRC2 inhibitor-treated macrophages[19]. Though attenuated macrophages displayed lower binding activity compared with virulent macrophages, PRC2 inhibition did not affect binding activity

(Fig. 4b). We injected the PRC2 inhibitor-treated macrophages into immune-compromised Rag2/γC mice and then orally administered the PRC2 inhibitor for 3 weeks to evaluate if loss of H3K27me3 impacts the formation of tumors in vivo. The parasite burden of heart and lung was quantified by measuring the amount of DNA corresponding to the *T. annulata ama-1* gene (TA02980), and all data are presented as mean ± SD. Similar to the fibronectin-binding assay, there was no significant change resulting from PRC2 inhibition (Fig. 4c).

### H3K27me3-mediated gene repression is dampened in attenuated macrophages
We next assessed gene expression changes following PRC2 inhibition in virulent and attenuated cells. For this purpose, we performed RNA-seq on *Theileria*-infected macrophages treated with UNC1999 or UNC2400 for 10 days to identify genes whose transcription is reactivated following loss of PRC2 activity. Consistent with the repressive function of H3K27me3, PRC2 inhibition resulted in a majority of differentially expressed genes (DEGs) being upregulated in both virulent and attenuated macrophages (Fig. 5a). Interestingly, the number of upregulated genes in virulent macrophages was more than 2-fold higher than in attenuated macrophages, suggesting that gene silencing is less dependent on PRC2 in attenuated macrophages. Twenty-one genes were commonly upregulated in virulent and attenuated macrophages, suggestive of a core function for H3K27me3-mediated repression in *Theileria*-infected macrophages (Fig. 5b, Supplementary Figs. 3, 4, and Supplementary Data 1). Gene ontology analysis also indicated that PRC2 retains its canonical function of regulating genes related to embryonic development in virulent macrophages. However, enrichment for this class of genes is no longer statistically significant in attenuated macrophages (Fig. 5c and Supplementary Data 2). This functional difference in PRC2 inhibitor responsiveness between virulent and attenuated macrophages could be caused by the difference in genome-wide distribution of H3K27me3 (Fig. 2b). Remarkably, some of the common PRC2-repressed genes (Fig. 5d) include Granzyme A (*GZMA*), identified as a novel dissemination suppressor of *Theileria*-transformed macrophages[35] and Follistatin (*FST*) known to reduce tumor invasiveness and metastasis in breast cancer[39,40] (Supplementary Fig. 3).

Overall, our analyses of the epigenetic landscape of *Theileria*-infected macrophages have uncovered a dramatic reconfiguration of the H3K27me3 landscape during the transition from the virulent to the attenuated state. The histone mark adopts a broad pattern of genomic distribution reminiscent of the transition from sharp to broad domains observed during development in the transition from the pluripotent to the differentiated state[41]. This is paralleled by a reduced transcriptional response following PRC2 inhibition, suggesting that this broad pattern may be less conducive to gene silencing and/or more resilient to PRC2 inhibition. Interestingly, silencing of some key tumor suppressor genes is preserved in attenuated macrophages, highlighting the requirement for a minimally functional PRC2 complex.

### Discussion
The unrestrained leukocyte proliferation and illicit dissemination within *Theileria*-infected animals have several similarities with human carcinogenesis, and for this reason, we focused on key post-translational modifications of histone tails with established links to oncogenic phenotypes. We chose to map H3K4me3 and H3K27me3 histone marks in virulent disseminating *T. annulata*-transformed versus attenuated macrophages with dampened dissemination[42]. As H3K27me3 is a key repressive mark, we hypothesized that changes in its genomic distribution could be involved in the attenuation processes. Indeed, we observed that the distribution of H3K27me3 is dramatically reconfigured during attenuation. In virulent macrophages, H3K27me3 preferentially accumulates around gene TSSs, while in attenuated macrophages the histone mark adopts a broader profile over large regions sometimes encompassing multiple genes (Figs. 2d and 3a, c). Furthermore, H3K27me3 peaks on the DEGs between virulent and attenuated macrophages also exhibited a similar distribution (Supplementary Fig. 5)[35].

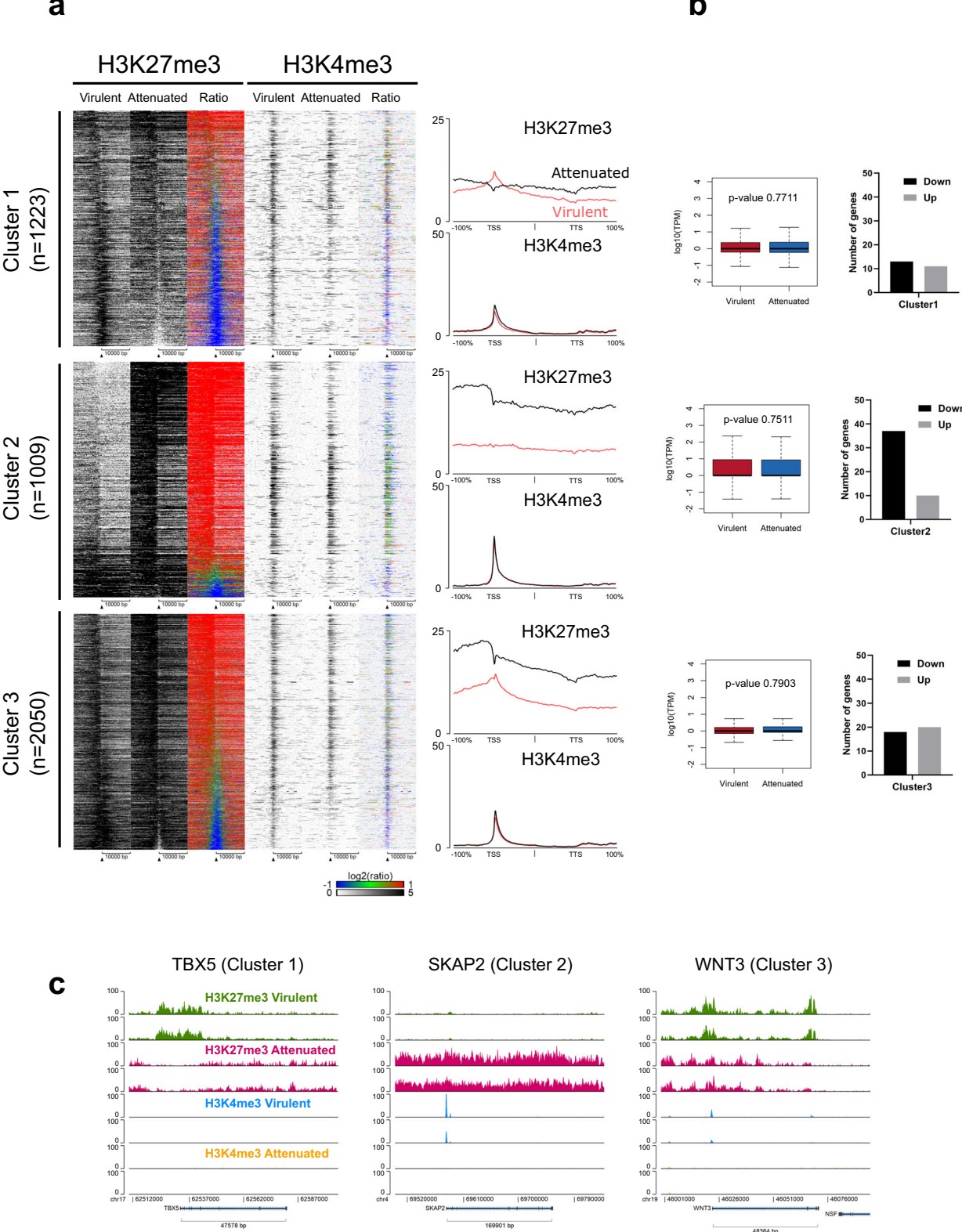

Given that follicular lymphoma harboring EZH2 mutations also displays a broadened H3K27me3 methylation profile, we looked at SNPs in both *ezh1* and *ezh2* genes of virulent and attenuated macrophages (data not shown), but could find no evidence of any amino acid changes. Long-term oxidative stress can also lead to increases in H3K27me3 levels, and attenuated macrophages exhibit elevated $H_2O_2$ production[43,44]. In fact, a number of different possible mechanisms could be involved, as reviewed in refs. [45,46]. For example, expression of PRC2 subunits and co-factors can increase relative to cell proliferation rate, but our RNA-seq data indicate that any differences are very subtle, as are changes in expression of H3K27me3 demethylases KDM6A (very slightly down) and KDM6B (very slightly up) in attenuated macrophages. Increased abundance in attenuated

**Fig. 3 | Association between shifts of H3K27me3 profile and gene expression.**
**a** H3K27me3 and H3K4me3 heat maps around TSS ± 10 kb of H3K27me3-positive genes either in virulent or attenuated macrophages. Genes were sorted by the log2-transformed ratio of H3K27me3 signal between attenuated and virulent macrophages (Att/Vir). Cluster 1: positive in virulent macrophages, Cluster 2: positive in attenuated macrophages, Cluster 3: positive in both macrophages. Replicate 2 heat maps are shown. Average H3K27me3 and H3K4me3 signal intensities of the genes of the corresponding clusters are shown on the right of each heat map. The *X*-axis represents a window centered on each gene, spanning from the transcription start site (TSS) to the transcription termination site (TTS) with an additional ±100%

flanking region. Red lines represent signals from virulent macrophages, whereas black lines represent those from attenuated macrophages. **b** Gene expression levels (box plots, left) and DEGs (histograms, right) of each cluster. Average TPM values of each cluster were compared between virulent and attenuated macrophages, and *p*-values were calculated using the Mann–Whitney *U* test. Within each cluster, the number of down or up DEGs in attenuated versus virulent macrophages was determined using DESeq2 (FDR < 0.1, *p*-value < 0.05, logFC > 1, logFC < −1).
**c** Genomic distribution of H3K27me3 and H3K4me3 around representative genes of each cluster.

**Fig. 4 | PRC2 inhibition does not alter the tumor dissemination properties of virulent or attenuated cells. a** PRC2 inhibitor treatment (UNC1999) lead to loss of the H3K27me3 histone mark, whereas the inactive analog UNC2400 had no detectable effect on H3K27me3 levels. Both macrophages were treated with DMSO or PRC2 inhibitors for 10 days, and nuclear extracts were prepared. **b** Fibronectin binding assay. Cells treated with PRC2 inhibitor were incubated in a fibronectin-coated well plate for 1 h at 37 °C. Error bars show ±SD from three biological replicates. For statistical analysis, two-way ANOVA was used. **c** Dissemination of *T. annulata*-infected macrophages in immune-compromised Rag2/γC mice. Gene copy number of parasites in each organ was obtained by absolute quantification of the *T. annulata ama-1* gene[80]. Wilcoxon rank-sum test was used for the analysis, and all data are presented as mean ± SD.

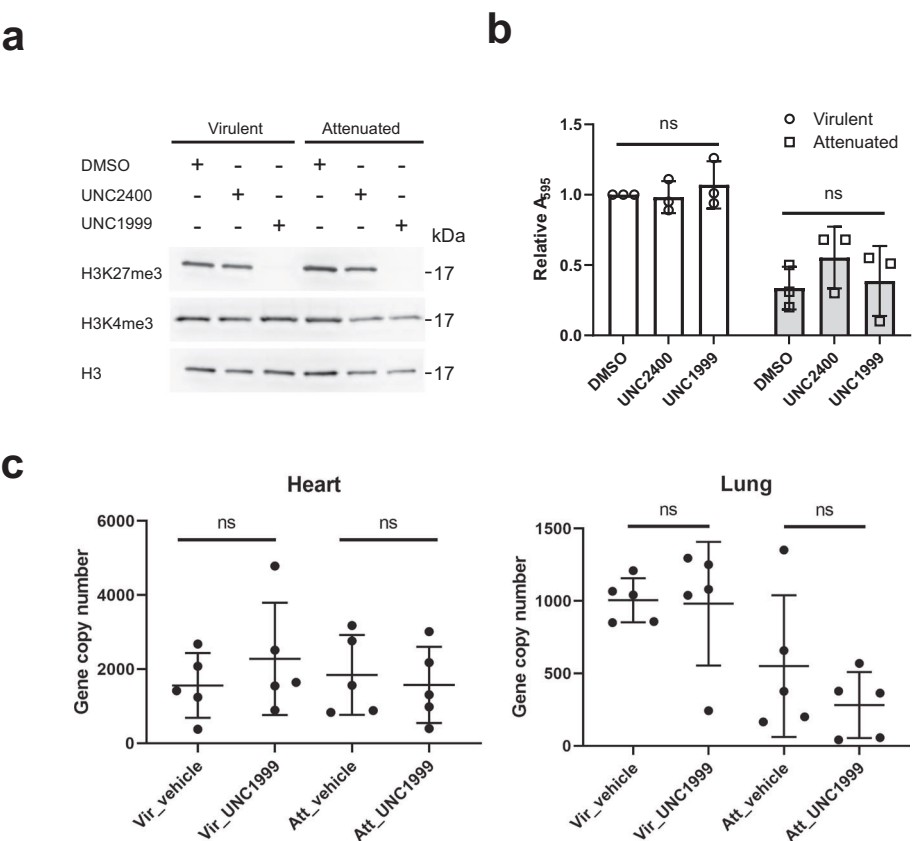

macrophages of H3K27ac or H3K36me3 marks might directly increase PRC2 activity, and decreased DNA methylation can also lead to an increase in PRC1 activity. Additionally, *T. annulata* is known to secrete a number of proteins predicted to be highly intrinsically disordered proteins (HIDP) such as TashAT2[47], Ta9[48], and the NIDP family[49], and all have been located to the host cell nucleus. Interestingly, in the context of chromatin silencing, overexpression of TashAT2 in BoMacs repressed transcription of 518 genes and upregulated transcription of 319 genes out of 837 DEGs[47]. Moreover, when Ta9 was overexpressed in BoMacs[50], the top-scoring transcription factor predicted to regulate Ta9-responsive DEGs was MTF2, a component of the PRC2 complex[51]. So, secreted parasite HIDPs might contribute to the formation of phase-separated compartments in the chromatin of host cells[52,53]. While these represent potential drivers of H3K27me3 spreading in attenuated macrophages, such spreading may arise from a combination of mechanisms.

Interestingly and importantly, we found that reconfiguration of H3K27me3 distribution is not associated with widespread gene silencing and that fewer genes become de-repressed following PRC2 inhibition in attenuated cells than in virulent macrophages. The 50 DEGs identified in attenuated macrophages through RNA-seq following PRC2 inhibitor treatment were integrated with ChIP-seq. Of these, eight genes (CD69, IRF1,

TMEM108, ENC1, CHN1, EHD3, GZMA, LY6E) were classified within cluster 2 in Fig. 3a, while the remaining genes were categorized into clusters 1 and 3. However, these analyses did not reveal any distinct chromatin features (Supplementary Fig. 4). These observations suggest that reconfiguration of H3K27me3 does not lead to a globally increased PRC2 silencing activity. Why spreading of H3K27me3 does not lead to a more extensive repression of transcription in attenuated macrophages is unclear, but we note that genes that gain H3K27me3 in attenuated macrophages are on average already lowly expressed in virulent cells (Fig. 3b). Previous studies have shown that global increases in H3K27me3 can lead to complex effects on transcription that do not necessarily result in a stronger repression of Polycomb target genes[54,55]. Interestingly, however, PRC2 inhibitor treatment led to the de-repression of three tumor suppressor genes, *GZMA, PRF1*, and *FST,* in both virulent and attenuated macrophages, indicating that H3K27me3 is required to maintain their repressed state, thus potentially contributing to the cancer-like phenotype of *Theileria*-induced transformation. This is consistent with our observation that *GZMA* plays a role in tumor dissemination of *Theileria*-infected leukocytes and human B-cell lymphomas[35]. The observation that genes silenced by PRC2 comprise both tumor suppressors and oncogenes may explain the lack of any obvious effect of inhibiting PRC2 on the transformed phenotype.

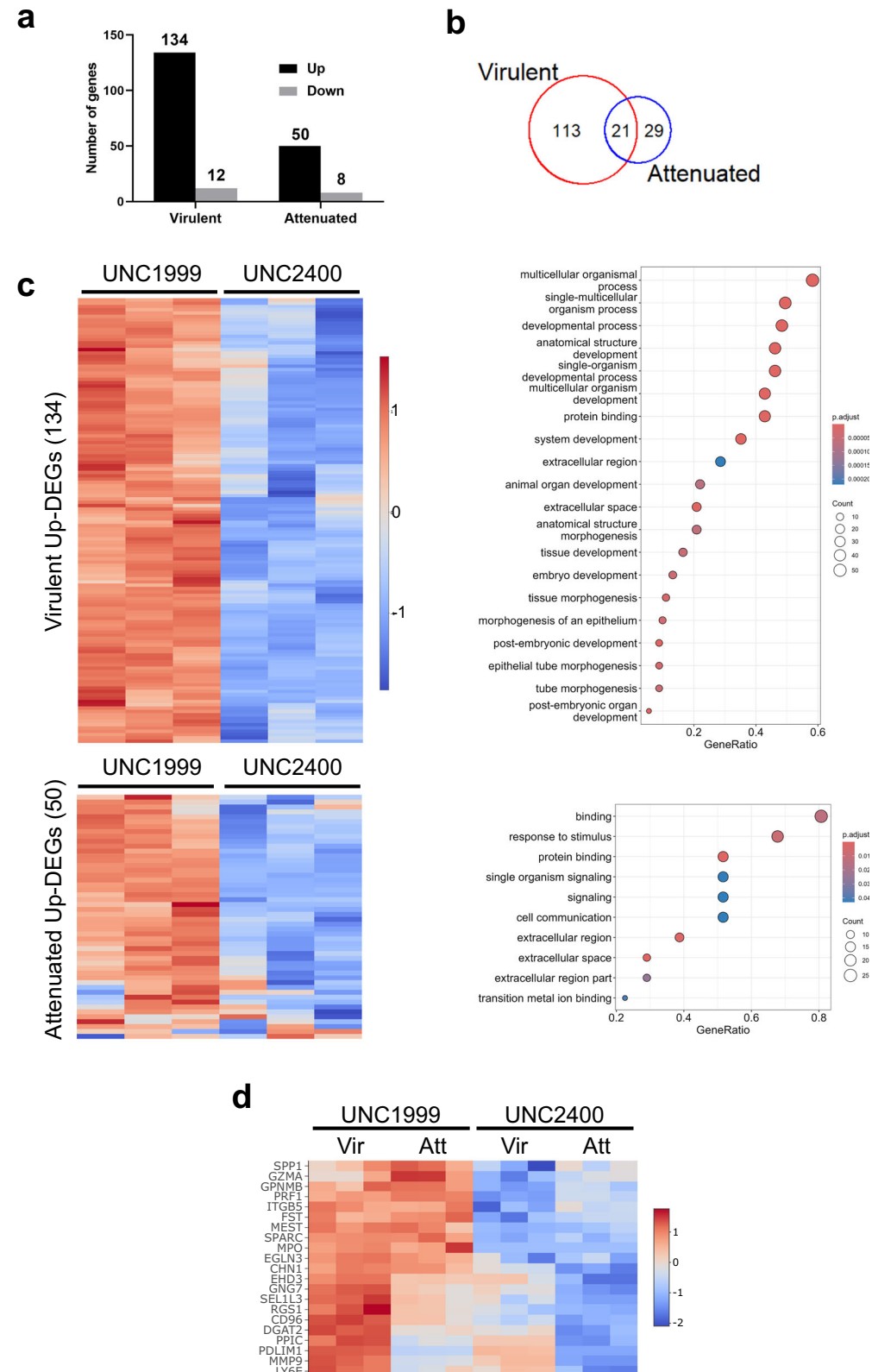

**Fig. 5 | H3K27me3-mediated gene repression is dampened in attenuated macrophages. a** Number of DEGs (logFC > 1, logFC < -1, padj < 0.05) between virulent and attenuated macrophages following PRC2 inhibitor treatment. **b** Venn diagram of common and uniquely upregulated genes in both virulent (134 genes) and attenuated (50 genes) macrophages. **c** Gene ontology analysis with the heat maps of upregulated DEGs. Gene ontology analysis was performed by Revigo[85] and clusterProfiler[86]. These pathways were specifically upregulated upon PRC2 inhibitor treatment in each macrophage. **d** Heat map of the 21 common upregulated genes in both macrophages upon UNC1999 treatment versus inactive analog UNC2400.

Alterations of the chromatin state following host-parasite infection have also been reported in other contexts. In particular, intracellular pathogens are capable of epigenetically modifying the gene expression of infected host cells by injecting effector molecules. *Legionella pneumophila* secretes the methyltransferase RomA into the host cell to modify H3K14 acetylation and promote intracellular bacterial replication[56]. Amastigotes of *Leishmania*, an obligate intracellular parasite, can induce epigenetic alterations of host macrophages, such as DNA and histone modifications to promote pathogen survival and replication[57]. For instance, modulation of host H3 acetylation and methylation by *Leishmania* amastigotes prevents NF-kB and NLRP3 inflammasome activation, resulting in suppression of host immune responses[58,59]. It is noteworthy that *Toxoplasma* E2F4-associated EZH2-inducing gene regulator (TEEGR) activates EZH2 and antagonizes NF-κB signaling by epigenetic repression in host cells[60]. Although host EZH2 protein levels appeared equivalent in attenuated macrophages (Fig. 2a), other mechanisms, such as phosphorylation[61,62] and oxidative stress[43,44], may contribute to heightened methyltransferase activity and/or decreased demethylase activity, thus resulting in spreading of the H3K27me3 mark in attenuated macrophages.

Evaluating H3K27me3 levels in non-infected leukocytes would be informative. However, uninfected primary macrophages do not proliferate unless stimulated and therefore would be difficult to use as a negative control for the analysis of H3K27me3 levels and genomic distribution. In fact, having a good negative uninfected control is a problem when studying *Theileria*-induced macrophage transformation, as it is more generally for cancer research, where the primary tumor progenitor is often not available. For *Theileria*-transformed macrophages, BoMacs are occasionally taken as a negative control[63], but BoMacs have been immortalized by the SV40 virus[64]. Thus, taking BoMacs as a negative control would entail comparing the H3K27me3 levels/distribution in SV40-immortalized macrophages with that observed in *Theileria*-transformed macrophages, which is far from ideal. Occasionally, in *Theileria* research, uninfected yet immortalized BL3 or BL20 cells are used as negative controls, especially when testing parasite-specificity of drug doses[65,66]. We compared H3K27me3 levels between BL3 and *T. annulata*-infected BL3 (TBL3), which revealed that infection is associated to a reduction in H3K27me3 levels (Supplementary Fig. 6a). However, as our focus was on determining if there was a role for PRC2-mediated silencing of virulence in attenuated Ode macrophages we reasoned that the best control to for attenuated Ode macrophages was to compare the distribution of H3K27me3 marks with that of virulent Ode macrophages, as they are isogenic and harbor the same *T. annulata* parasite strain. To that end, virulent *T. annulata*-transformed macrophages were treated with buparvaquone (BPQ), following the protocol described in ref. [67], to selectively eliminate the parasite without affecting host cell viability. Total RNA was extracted, and the expression levels of bovine *ezh1* and *ezh2*, as well as parasite *p104* (a control for parasite clearance), were assessed by qRT-PCR (Supplementary Fig. 6b). *Theileria* infection appears to lead to a downregulation of both *ezh1* and *ezh2*, and this may be a first step in infection-induced macrophage dedifferentiation[68,69]. Moreover, it might also enable parasite infection to rewire host cell gene expression[70] and the increase in H3K4me3 signal (at the exposure used) is consistent with rewiring of the host transcriptome.

Our roadmap for future research, however, does not include further characterization of PRC2-mediated H3K27me3 silencing, since our in vivo mice study suggests that inhibiting PRC2 activity is not a clinically viable control strategy (Fig. 4). PRC2-inhibitor treatment had no effect on *Theileria*-induced tumor dissemination in Rag2γC mice, and its application to treat cattle is difficult to imagine given the large quantities required. Rather, we propose to concentrate on other repressive marks that could potentially play a role in the loss of *Theileria*-transformed macrophage virulence, with a particular interest in H3K9me3[71] for its potential role in repressing the expression of specific host and/or parasite virulence genes[72].

## Methods

### Cell culture

*Theileria annulata*-infected macrophages are the Ode virulent and attenuated lines corresponding to passages 53 and 309[29,73]. These cells were maintained with RPMI medium supplemented with 10% fetal bovine serum (FBS), 2 mM L-glutamine, 100 U penicillin, 0.1 mg/ml streptomycin, and 4-(2-hydroxyethyl)-1-piperazineethanesulfonic acid (HEPES) at 37 °C with 5% $CO_2$.

### Immunofluorescence microscopy

$1 \times 10^5$ cells were centrifuged on a glass slide with the Cellspin I (Tharmac) at 1500 rpm for 3 min and fixed with 4% paraformaldehyde for 10–15 min at room temperature. Fixed cells were permeabilized with 0.2% Triton X-100 for 10 min and blocked with 1% BSA for 30 min. These cells were incubated with primary antibodies against H3K27me3 (1/1000, Cell Signaling Technology, #9733) or H3K4me3 (1/1000, Cell Signaling Technology, #9751) with parasite surface protein p104 (1/1000, 1C12)[74] for 1 h at room temperature, sequentially stained with secondary antibodies conjugated with Alexa 488 and Alexa 594 (1/1000, Molecular Probes) for 45 min at room temperature. Stained cells were mounted in ProLong Diamond Antifade Mountant with DAPI (Thermo Fisher Scientific). Stained cells were imaged by a wide-field Leica DMI6000, and these images were deconvoluted.

### Preparation of nuclear extracts

We washed $0.5–2 \times 10^7$ *T. annulata*-infected macrophages, and *D. melanogaster* S2 cells with phosphate-buffered saline (PBS) once and subsequently suspended them with Buffer A (10 mM pH 7.9 HEPES, 5 mM MgCl$_2$, 250 mM sucrose, 0.1% NP40) supplemented with protease inhibitors and 1 mM DTT. After keeping on ice for 10 min, cell suspensions were centrifuged at 8000 rpm for 10 min at 4 °C to collect the cytosolic fraction. Pellets were subsequently suspended with Buffer B (25 mM pH 7.9 HEPES, 1.5 mM MgCl$_2$, 0.1 mM EDTA, 20% Glycerol, 700 mM NaCl) supplemented with protease inhibitors and 1 mM DTT, followed by sonication for 10 min on ice. Following centrifugation (14,000 rpm for 15 min at 4 °C), nuclear fractions were collected from the supernatant. Collected nuclear extracts were quantified using the Bradford protein assay.

### Immunoblotting

Nuclear extracts were run by SDS–PAGE and transferred to a nitrocellulose membrane (GE Healthcare). Subsequently, membranes were blocked with 5% (w/v) skimmed milk powder in PBS containing 0.1% Tween 20 (PBS-T) for 1 h at room temperature. Immune reactions were carried out using 1/2000 diluted monoclonal rabbit anti-H3K27me3 or H3K4me3 antibodies (Cell Signaling Technology), 1/1000–1/3000 diluted monoclonal mouse H3K27me1 antibodies (Active Motif, #61015), 1/2000 diluted monoclonal mouse H3K27me2 antibodies (Active Motif, #61435), 1/3000 diluted rabbit polyclonal EZH2 antibodies (homemade) or 1/50,000 diluted monoclonal rabbit anti-H3 antibodies (Active Motif, #39163) overnight at 4 °C as primary antibody. 1/5000 diluted HRP-labeled anti-rabbit secondary antibodies (Santa Cruz) for 1 h at room temperature. Proteins were visualized with X-ray film or ECL (Thermo Fisher Scientific) and imaged with fusion FX (Vilber Lourmat). The H3 levels were used as a loading control, and unedited and uncropped blots are presented in Supplementary Information 5.

### Preparation of chromatin

We cross-linked $1.5 \times 10^7$ cells by pre-warmed DMEM supplemented with 1% Formaldehyde, 15 mM NaCl, 150 μM EDTA, 75 μM EGTA, and 15 mM pH 8.0 Hepes for 10 min at room temperature with slow agitation and then quenched formaldehyde with 125 mM Glycine for 5 min at room temperature. Following centrifugation at 1000 rpm for 10 min at 4 °C, cells were washed with cold PBS once. Washed cells were suspended with 1 ml Buffer 1 (50 mM pH 7.5 Hepes–KOH, 140 mM NaCl, 1 mM EDTA, 10% glycerol, 0.5% NP40, 0.25% Triton X-100) supplemented with protease inhibitors

(PMSF, aprotinin, leupeptin, pepstatin) and rocked at 4 °C for 10 min. Following centrifugation at 1500 rpm for 5 min at 4 °C, cells were resuspended with 1 ml Buffer 2 (10 mM pH 8 Tris, 200 mM NaCl, 1 mM EDTA, 0.5 mM EGTA) supplemented with protease inhibitors (PMSF, aprotinin, leupeptin, pepstatin) and rocked at room temperature for 10 min. Cells were centrifuged at 1500 rpm for 5 min at 4 °C and resuspended 1.3 ml Buffer 3 (10 mM pH 8 Tris, 1 mM EDTA, 0.5 mM EGTA, 0.5% N-lauroyl-sarcosine) supplemented with protease inhibitors (PMSF, aprotinin, leupeptin, pepstatin). Suspended cells were sonicated using the Bioruptor (Diagenode) with the setting (high, 30 s intervals for 30 min) and chromatin was collected after centrifuge at 14,000 rpm for 10 min at 4 °C. The size of reverse cross-linked DNA was 300–500 bp determined by DNA electrophoresis. Each chromatin sample was quick-frozen in liquid nitrogen and stored at −80 °C.

### Chromatin-DNA quantitation

We mixed 20 μl of chromatin with 180 μl of $T_{50}E_{10}S_1$ (50 mM Tris pH 8.0, 10 mM EDTA, 1% SDS) at 65 °C overnight for reverse-crosslinking. We added RNase A to chromatin samples and then incubated them for 1 h at 37 °C to eliminate RNA from the samples. Samples were incubated with Proteinase K for 1 h at 55 °C. Extract samples were mixed with phenol:chloroform:isoamyl-alcohol and sequentially added 20 μg glycogen. Purified chromatin-DNA was collected by ethanol/NaCl precipitation, and its concentration was determined with a Qubit Fluorometer (Thermo Fisher Scientific).

### Chromatin-immunoprecipitation (ChIP)

Dynal magnetic beads (Thermo Dynabeads Protein A: 10001D) were incubated with 20 μl of H3K4me3 or H3K27me3 antibodies for 8 h at 4 °C. We prepared chromatin solution mixed with 20 or 10 μg chromatin for H3K4me3 or H3K27me3 antibodies with 5% S2 chromatin in incubation buffer (3% Triton X-100, 0.3% sodium deoxycholate, 15 mM EDTA) supplemented with protease inhibitors (PMSF, aprotinin, leupeptin, pepstatin). We combined the chromatin with antibody-coupled beads and rotated the mixture overnight at 4 °C. Each bead was washed 6 times by ice-cold RIPA (50 mM Hepes–KOH pH 7.5, 10 mM EDTA, 0.7% sodium deoxycholate, 1% NP-40, 0.5 M LiCl) supplemented with protease inhibitors (PMSF, aprotinin, leupeptin, pepstatin) and additionally washed Tris-buffer (10 mM Tris pH 8.0, 1 mM EDTA, 50 mM NaCl). We eluted chromatin with 200 μl of $T_{50}E_{10}S_1$ (50 mM Tris pH 8.0, 10 mM EDTA, 1% SDS) at 65 °C for 30 min by shaking at maximum speed. Beads were centrifuged at 14,000 rpm for 1 min and placed on the magnetic stand. After revers-crosslinking described above, purified chromatin-DNA concentration was determined with a Qubit Fluorometer (Thermo Fisher Scientific).

### ChIP-Seq analysis

We mapped all precipitated reads to *Bos taurus* (bosTau8) and *D. melanogaster* (dmel5.41) genomes by bowtie2-2.1.0 and processed the generated files by Samtools[75,76]. Normalization factors were calculated based on the number of reads mapped to the *D. melanogaster* spike-in genome (Supplementary Data 3)[34]. As we expected, around 90% reads precipitated by the H3K27me3 antibody were mapped to the *B. taurus* genome, while H3K4me3 were mapped to a relatively low percentage (32.18–63.12%). This is because H3K4me3 antibodies successfully precipitated marks of both *T. annulata* and *B. taurus* histone 3 tails, and the unmapped reads to either *B. taurus* or *D. melanogaster* genomes were mapped to the *T. annulata* genome (data not shown). Peak calling was performed using adaptive local threshold (ALT) equipped in Easeq (windowsize: 3000 bp for H3K27me3, 1500 bp for H3K4me3, p-value: 1E−5, false discovery rate (FDR): 1E−5, Log2fold diff: 2, merge within: 100 bp) or MACS2 algorithm and subsequently annotated to *B. taurus* transcripts (UMD3.1.1, n = 44,681)[77,78]. For clustering of each ChIP-seq, we performed MACS2 peak call and clustering of all data by Bioconductor DiffBind package[79]. DiffBind clustering of peaks and a high Pearson's correlation coefficient (r > 0.95) between two biological replicates of each macrophage and antibody indicated that the ChIP-seq had worked correctly (Supplementary Fig. 2b and c). We calculated the H3K27me3 and

H3K4me3 ratio of the quantified value of each gene around ±2 kb from TSS between virulent and attenuated cells. Venn diagrams of the histone mark positive transcripts between virulent and attenuated macrophages were generated by R.

### Fibronectin binding assay

Virulent and attenuated macrophages were treated with DMSO, 1 μM UNC1999, or 1 μM UNC2400 (inactive analog) for 10 days[37,38]. We seeded treated macrophages onto fibronectin-coated plates and incubated them for 1 h at 37 °C. Bound cells after PBS wash were stained with Violet blue and lysed by the lysis buffer containing Triton X-100. The number of cells was quantified by measuring absorbance at 595 nm.

### Dissemination of *T. annulata*-transformed macrophages in immune-deficient mice

*T. annulata*-transformed virulent and attenuated macrophage lines were treated in vitro for 10 days with DMSO or 1 μM UNC1999, as described above. The macrophages ($10^6$ in 200 μl PBS) were then injected subcutaneously into the right flank region of 'alymphoid' Rag2/γC double-deficient (T-, B-, NK-) mice and surveyed for 3 weeks[35]. Four groups, each containing five age and sex matched mice, were defined: two treatment groups, including virulent/attenuated Ode treated with EZH2 inhibitor, and two control groups, including virulent/attenuated Ode treated only with drug vehicle (solvent). During the 3-week study period, mice in the treatment groups were orally administered with UNC1999 (100 mg/kg/day) dissolved in a solution of Tween® 80 (Sigma-Aldrich, Germany) and carboxymethyl cellulose sodium salt (Sigma Life Science, USA) to facilitate oral gavage. The control mice received 100 μl of solvent per day. At the end of the study, mice were humanely sacrificed, and organs were dissected and kept in PBS at −20 °C.

### Absolute quantification of *T. annulata*-infected cells in mice tissues

A single copy *T. annulata* gene (apical merozoite antigen 1, TA02980) was cloned into pJET 1.2/blunt cloning vector using CloneJET PCR Cloning Kit (Thermo Fisher Scientific). To measure the load of *T. annulata*-infected macrophages in each tissue, quantitative PCR technology was applied to genomic DNA extracted from mouse tissue (QIAmp DNA mini kit). Quantification was done based on the methodology described elsewhere[80]. Briefly, a standard curve was generated from crossing points (Cp) measured from serial dilutions of pJET-ama plasmid construct with known plasmid copy numbers using a qPCR assay. The Cps from 1 ng DNA of each tissue sample were used to estimate parasite gene copy numbers based on the equation derived from the plotted data of the standard curve graph.

### Library preparation and sequencing

We maintained virulent and attenuated macrophages with 1 μM PRC2 inhibitor for 10 days, after which a total of $2.5 \times 10^6$ cells were collected for RNA extraction using mirVana miRNA isolation Kit (Thermo Fisher Scientific, catalog number AM1560) according to the manufacturer's protocol. Strand-specific RNA-sequencing (ssRNA-seq) libraries were prepared using the Illumina Truseq Stranded mRNA Sample Preparation Kit (Illumina, catalog number RS-122-2101) following the manufacturer's instructions. Briefly, 1 μg of total RNA was used to purify mRNA using poly-T oligo-attached magnetic beads. mRNA was then fragmented, and cDNA was synthesized using SuperScript III reverse transcriptase (Thermo Fisher Scientific, catalog number 18080044), followed by adenylation on the 3' end, barcoding, and adapter ligation. The adapter-ligated cDNA fragments were then enriched and cleaned with Agencourt Ampure XP beads (Agencourt, catalog number A63880). Library validation was conducted using the 1000 DNA kit on 2100 Bioanalyzer (Agilent Technologies, catalog number 5067-1504) and quantified using Qubit (Thermo Fisher Scientific, catalog number Q32850). ssRNA libraries were sequenced on Illumina Hiseq 4000.

## RNA-seq analysis

All reads were trimmed using Trimmomatic and then aligned to the *Bos taurus* UMD.3.1 genome with HISAT2[81]. BAM files converted by Samtools, as described above and were processed by HTSeq-count tools[82] to quantify mapped reads on each gene. We obtained a DEGs list among each group with logFC > ±1 and padj < 0.05 using EdgeR and Limma of NetworkAnalyst or TCC-GUI[83,84] separately, and made intersections of these two lists to identify more stringent DEGs. Gene ontology analysis was performed by Revigo[85] and clusterProfiler[86].

## Gene expression measurements by qRT-PCR

Total RNA was extracted from cells using RNAeasy® plus mini kit (QIAGEN). Complementary DNA (cDNA) synthesis was done by using 1 μg RNA as template and the GoScript™ reverse transcriptase kit (Promega, Cat. No. A5001, USA), in 20 μl reaction volume, following the manufacturer's instructions. To perform qRT-PCR, target sequences were amplified in a SYBR green PCR master mix (ThermoFisher Scientific, Cat. No. 4309155) mixed with diluted cDNA (1:20), double-distilled water and primers (Supplementary Data 4). The PCR was run in the LightCycler® 480 instrument (Roche) and the results were analyzed in Microsoft Excel. Finally, the $2^{-\Delta\Delta CT}$ methodology was employed to estimate relative gene expression levels[87]. The expression of Bovine GAPDH was used as the internal control for normalization. Graphs were prepared in GraphPad Prism v8.4.0.

## Ethics statement

A detailed protocol (number 12-26) describing the above mice experiments was first submitted to and approved (number CEEA34.GL.03312) by the ethics committee for animal experimentation at the University of Paris-Descartes. The university ethics committee is registered with the French National Ethics Committee for Animal Experimentation, which itself is registered with the European Ethics Committee for Animal Experimentation. The right to perform the mouse experiments was obtained from the French National Service for the Protection of Animal Health and satisfied the animal welfare conditions defined by laws (R214-87–R214-122 and R215-10), and G.L. was responsible for all experiments as he holds the French National Animal Experimentation permit with the authorization number (B-75-1249). All experimental procedures in KAUST were approved by the Institutional Biosafety and Bioethics Committee (IBEC) in KAUST (IBEC number: 22IBEC029). We have complied with all relevant ethical regulations for animal use.

## Statistics and data reproducibility

ChIP-seq was performed using two independent biological replicates (Supplementary Fig. 2). RNA-seq of PRC2 inhibitor-treated cells was performed in biological triplicate. The two-group comparisons were performed using the Wilcoxon rank-sum test (Mann-Whitney *U* test), and multiple-group comparisons were performed using two-way ANOVA. Statistical analyses were performed using GraphPad Prism (v8.4.0) or R (v4.0.2, v4.1.0).

## Reporting summary

Further information on research design is available in the Nature Portfolio Reporting Summary linked to this article.

## Data availability

All raw data of ChIP-seq and RNA-seq have been deposited in the European Nucleotide Archive and are publicly available (https://www.ebi.ac.uk/ena/browser/view/PRJEB52136). Source data are provided in Supplementary Data 5.

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

## Acknowledgements

T.S. and S.T. acknowledge a LabEx ParaFrap postdoctoral fellowship, and G.L. acknowledges ANR-11-LABX-0024 support and core funding from INSERM and the CNRS. G.L. and A.P. acknowledge financial support received from KAUST in the form of a joint (OCRF-20146CRG4) grant and A.P. a BAS/1/1020-01-01 grant. We thank Raphael Margueron at the Curie Institute for his assistance with the ChIP-seq experiments; the Next Generation Sequencing Platform at the Curie Institute for ChIP sequencing, and members of the KAUST Bioscience Core Laboratory (BCL) for generating the raw RNA-seq datasets. We also thank Axel Martinelli at Hokkaido University for instruction on RNA-seq analysis and Mads Lerdrup at the University of Copenhagen for guidance on ChIP-seq analysis by EaSeq. We thank Brian Shiels at Glasgow University for the gift of the monoclonal antibody (1C12) specific for the p104 antigen.

## Author contributions

T.S. performed all wet-bench experiments, ChIP-seq bioinformatic analyses, and wrote the manuscript with editorial inputs from A.P., M.W., and G.L. S.T. performed all mouse experiments, including estimation of tumor dissemination and qRT-PCR analysis. He also did the wet-bench experiments requested by the reviewers. Z.R. made the RNA-seq libraries that were bioinformatically analyzed by T.S., H.R.A., A.K., and T.M. The study was conceived by M.W. and G.L., and supported by a joint grant awarded to A.P. and G.L.

## Competing interests

The authors declare no competing interests.
