## [Transparent Peer Review file · Communications Biology]

Epigenetic landscape of the H3K27me3 mark in macrophages transformed by *Theileria annulata*

Corresponding Author: Dr Gordon Langsley

Version 0:

Reviewer comments:

Reviewer #1

(Remarks to the Author)

The study addresses an important and timely question in the fields of parasitology, epigenetics, and cancer biology. It explores how *Theileria annulata* infection alters the epigenetic landscape of host macrophages, focusing on the H3K27me3 mark, and provides valuable insights into the molecular changes associated with the attenuation of virulence. The findings have potential implications for understanding both parasite-induced leukocyte transformation and broader mechanisms of oncogenesis. However, there are areas where the study could be strengthened to enhance its impact and clarity. Below are comments to improve the paper:

What happens to the expression of EED and SUZ12 the other two core subunits of Polycomb repressive complex 2 (PRC2) other than the EZH2, which plays pivotal roles in transcriptional regulation. Expression of these two EED and SUZ12 should be checked in attenuated and virulent lines after the treatment with PRC2 inhibitor.

The fact that attenuated macrophages have a low dissemination potential led us to hypothesize that oncogenes activated as a consequence of *Theileria* infection might be transcriptionally downregulated through mechanisms promoting a repressed chromatin state. In the end of the study have you find any oncogene linked to the H3K27me3 distribution in virulent lines.

While the paper identifies changes in H3K27me3 distribution and links them to PHF19 expression, the mechanistic connection between PHF19 reduction and H3K27me3 spreading remains speculative. If possible Include experimental validation, such as PHF19 knockdown in virulent macrophages, to test whether this alone can recapitulate the H3K27me3 redistribution observed in attenuated cells.

The study notes that H3K27me3 redistribution does not lead to widespread gene silencing but does not fully explain why this is the case.

The paper identifies several unanswered questions (e.g., the role of *Theileria* in modulating H3K27me3, the impact of PHF19 reduction) but does not propose a clear roadmap for future research. Include a dedicated "Future Directions" section outlining specific experiments and approaches to address these questions.

briefly address the potential clinical relevance of its findings. For instance, could targeting PRC2 or PHF19 be a viable strategy for controlling *Theileria* infections or related cancers?

While the study identifies changes in H3K27me3 distribution, it does not fully elucidate the underlying mechanisms. For example, what drives the broadening of H3K27me3 in attenuated macrophages?

The study mentions that H3K27me3 maintains the silencing of key tumor suppressor genes but does not elaborate on the functional consequences of this silencing. Suggestion: Discuss more about how the silencing of these genes contributes to the transformed phenotype of *Theileria*-infected macrophages and whether this has implications for the parasite's survival or dissemination.

Reviewer #2

(Remarks to the Author)

In this manuscript, Sakura et al investigates the dynamics of the H3K27me3 chromatin mark during long-term passage of bovine macrophages infected with *T. annulata*. Through immunofluorescence staining (Fig. 1), Western blot (Fig. 2A) and ChIP-Seq (Fig. 2C-D), they observe that attenuated macrophages display an increase and altered pattern of H3K27me3 deposition vs virulent macrophages. Whereas in the latter, H3K27me3 is enriched at transcription start sites (TSS), in the former H3K27me3 was most enriched in intergenic regions. However, at a global level by RNA-Seq (Fig. 3), there were no statistical differences in expression of genes in virulent vs attenuated macrophages when sub-setted on different patterns of H3K27me3 enrichment. Although H3K27me3 is enriched in attenuated macrophages, pharmacological inhibition of PRC2 minimally de-represses gene expression in attenuated macrophages (in fact, PRC2 inhibition results in greater transcriptional de-repression in virulent vs attenuated macrophages) (Fig. 5) and does not rescue the invasive phenotype of attenuated macrophages in *in vitro* and *in vivo* assays (Fig. 4). The authors point to recently identified functional tumor suppressors (GZMA, FST) as PRC2-repressed genes (Fig. 5).

The authors identify interesting chromatin dynamics during the transition from virulent to attenuated phenotype of infected macrophages, yet whether these changes are functionally important for macrophage attenuation is unclear. Nevertheless the manuscript could be strengthened in a few key areas:

- 1) Inclusion of uninfected control. Particularly for analysis of H3K27me3 levels and distribution by ChIP-Seq (Fig. 2), it would be informative to understand how H3K27me3 looks in uninfected cells. This would provide greater context for the H3K27me3 changes studied here between virulent and infected macrophages by establishing a baseline.
- 2) Quantification of immunofluorescent images. The increase in H3K27me3 levels in bulk attenuated macrophages is quite clear (Fig. 2). However the immunofluorescent images shown in Fig. 1B present a unique opportunity to extend the analysis. At a single-cell level, how do H3K27me3 levels look in attenuated vs virulent macrophages? Is this mark uniformly increased in all attenuated macrophages or certain subsets? And is there any correlation with H3K27me3 levels and parasite burden (p104 staining)?
- 3) Further integration of ChIP-Seq and RNA-Seq data. In Fig. 3, the authors note a small number of genes which have lower expression in attenuated macrophages coinciding with the increased H3K27me3 in these cells. Upon further inspection of SKAP2, which is downregulated in expression in attenuated macrophages, this gene locus not only gains H3K27me3 but also loses H3K4me3 (Fig. 3C). H3K4me3 levels don't appear to be globally different in virulent vs attenuated macrophages (Fig. 3B) but are DEGs characterized by gain of H3K27me3 AND loss of H3K4me3? Furthermore, is the RNA-Seq data used here generated by the authors or from the literature? This should be made more clear
In a similar fashion, are there any defining chromatin features of genes subject to PRC2-mediated repression in Fig. 5? Integration of these data with ChIP-seq could yield some insights.
- 4) Readout of *in vivo* assay. In Fig 4, the authors look at parasite burden in heart and lung tissues as their readout after PRC2 inhibition in attenuated macrophages. Did the authors look at engraftment or dissemination of the macrophages themselves?
- 5) Connection with TGF- β . The authors cite that TGF- β signaling is a known signal that can restore the dissemination capacity of attenuated macrophages, suggesting that epigenetic changes may contribute to the attenuated phenotype. Can the authors expand on why they believe TGF β may be working via the epigenome? Further yet, it could be interesting for the authors to look at H3K4me3 and H3K27me3 in attenuated macrophages treated with TGF β . Regardless of the result, it could shed some light on the mechanisms underlying TGF β in establishing the attenuated macrophage phenotype and the importance of epigenetics in this TGF β -mediated process.

Minor comments:

- 1) Fig. 5C is a very untraditional way to display pathway analysis. Please either perform gene set enrichment (GSEA) or show p-values for top pathways.
- 2) In some RNA-Seq legends and ChIP-Seq tracks, the axes labels are quite small and could be increased in size to enhance readability.

Reviewer #3

(Remarks to the Author)

The authors have attempted to look at the molecular events that underpin the attenuation of *Theileria annulata* infected cells. Specifically, they've looked at H3K27me3 and H3K4me3 modification profiles of pathogenic vs attenuated cells. Generally, the interesting observation is the re-distribution of H3K27me3 from tight peaks at TSS in pathogenic to spread around TSS in attenuated cells. Although the biological significance of this observation, in the context of the pathogenic vs attenuated phenotype, has not been demonstrated considering that they've shown it not due to significant changes in gene expression. The work appears straightforward, except it would have been nice to either show, or provide reference, phenotypic evidence that the cells are indeed pathogenic or attenuated. While there's RNA-seq analysis on PRC2-inhibitor treated cells, it would have been good to show expression levels of genes like SKAP2 that show the changes in H3K17me3 distribution in attenuated cells. There is a reference to a supplementary table 2 with expression levels, but I didn't see it. There discussion seem to introduce new results that should be included in the results section rather than discussion e.g., line 211-214. Line 155 has (ref), which I assume was a placeholder for the author to include a reference. As there's no statistical test, the westernblot results on Fig2A, should be described as numerical higher on line 114

Version 1:

Reviewer comments:

Reviewer #1

(Remarks to the Author)

The edits look good and have fully resolved my queries. I am satisfied with the updated manuscript.

Reviewer #2

(Remarks to the Author)

This reviewer appreciates the efforts of the authors in addressing all comments, particularly the lengths to which they went to address point #1.

This reviewer would like to make two small requests: i) that single-cell quantification of H3K27me3 levels in virulent vs attenuated cells be added to Fig. 1 and/or its supplement; and ii) to include in the main figure 5c (not just supplement) p-values corresponding to the listed pathways in the pathway analysis as its presentation is currently unscientific.

Reviewer #3

(Remarks to the Author)

The authors have responded to the reviewers comments comprehensively and performed additional analyses to support their claim of differential H3K27m3 distribution in pathogenic and attenuated *T. annulata* infected cells.

Editor's comments

1 – Please address point 1 from reviewer #2.

“Inclusion of uninfected control. Particularly for analysis of H3K27me3 levels and distribution by ChIP-Seq (Fig. 2), it would be informative to understand how H3K27me looks in uninfected cells. This would provide greater context for the H3K27m3 changes studied here between virulent and infected macrophages by establishing a baseline.”

We have analysed *ezh1/ezh2* expression in drug-cured, parasite-free, macrophages and to our surprise found that infection down-regulated their expression. We confirmed their down-regulation by our own bioinformatic analysis of publicly accessible RNA-seq raw reads (ENA accession code: PRJEB65111) derived from macrophages infected *in vitro* by *T. annulata* sporozoites. Since the raw read data was not generated by us and has not yet been published, we present our bioinformatic analysis uniquely in our rebuttal (see replies to Reviewer #2). This showed (highlighted in yellow) that several PRC2 complex genes display significantly downregulated expression 3-weeks post-infection.

In our reply to Reviewer #2 we confirmed infection-induced down-regulated expression of *ezh1/ezh2* at the protein level by western blot using an alternative negative control model; namely, BL3 B cells compared to infected BL3 (TBL3) cells.

In spite of the originality of these new data we have not integrated the infection-related data into our revised manuscript, whose focus was/is to determine if the alterations in the distribution of H3K27me3 repressive marks contributed to diminished dissemination of attenuated macrophages. However, we have integrated a discussion of them into “Future Directions” in the Discussion, as requested by Reviewer #1. We discuss how infection-induced reduced levels of H3K27me3 could be a first step in infection-induced macrophage dedifferentiation, and in addition, it might also enable parasite infection to rewire host cell gene expression.

2 – As requested by reviewer #2 (point 3), please specifically state whether the RNAseq data was generated by you or whether the dataset was retrieved from the existing literature “Is the RNA-Seq data used here generated by the authors or from the literature? This should be made more clear.”

In our reply to Reviewer #2 we pointed out that the RNA-seq data on virulent and attenuated Ode macrophages was generated by us and cited in the original submission as reference 35.

Reviewer #1 (Remarks to the Author)

The study addresses an important and timely question in the fields of parasitology, epigenetics, and cancer biology. It explores how *Theileria annulata* infection alters the epigenetic landscape of host macrophages, focusing on the H3K27me3 mark, and provides valuable insights into the molecular changes associated with the attenuation of virulence. The findings have potential implications for understanding both parasite-induced leukocyte transformation and broader mechanisms of oncogenesis.

We appreciate the very positive comment, “provides valuable insights with potential implications for understanding both parasite-induced leukocyte transformation and broader mechanisms of oncogenesis.”

However, there are areas where the study could be strengthened to enhance its impact and clarity. Below are comments to improve the paper:

What happens to the expression of EED and SUZ12 the other two core subunits of Polycomb repressive complex 2 (PRC2) other than the EZH2, which plays pivotal roles in transcriptional regulation. Expression of these two EED and SUZ12 should be checked in attenuated and virulent lines after the treatment with PRC2 inhibitor.

The RNA-seq data that we quoted on virulent and attenuated Ode infected macrophages (the same as used in the current study) was produced and published by us (cited as reference 35 in the manuscript). From these datasets we pulled PRC2-associated genes and, as requested by Reviewer #1 show below that EED and SUZ12 displayed no significant change in expression between virulent and attenuated Ode macrophages.

RNA-seq data on PRC2-associated EED and SUZ12 genes expressed in virulent & attenuated macrophages

gene_name	gene_id	sample_1	sample_2	value_1	value_2	log2_fold_change	test_stat	p_value	q_value	significant
SUZ12	gene21359	ODE_VIR	ODE_ATT	58,4431	54,2367	-0,107763	-0,659489	0,24635	0,551792	no
EED	gene28636	ODE_VIR	ODE_ATT	45,7524	43,3632	-0,0773763	-0,422145	0,46585	0,751383	no

The fact that attenuated macrophages have a low dissemination potential led us to hypothesize that oncogenes activated as a consequence of *Theileria* infection might be transcriptionally downregulated through mechanisms promoting a repressed chromatin state. In the end of the study have you find any oncogene linked to the H3K27me3 distribution in virulent lines

To satisfy the reviewer’s question we compared to the COSMIC (cosmickb.org) database the 113 genes in virulent macrophages that displayed upregulated expression upon PRC inhibitor treatment (Table S2) i.e., repressed by PRC2-mediated K27me3 marks and identified 5 genes (see below).

PTPRK expression is described on NCBI as stimulated by TGF- β and CHST11 and CTNND1 figured on the list of 76 potential TGF-target genes we previously published (PMID: 21124992). PTPRK was not listed in this study, as its corresponding oligomers were not printed on the microarray used in 2010 (PMID: 21124992). We would have expected CHST11 and CTNND1 to be strongly expressed due to high TGF β 2 levels in virulent Ode (PMID: 21124992), but their expression appears repressed by PRC2-mediated H3K27me3 chromatin marks, and only increases upon PRC2 inhibition.

While the paper identifies changes in H3K27me3 distribution and links them to PHF19 expression, the mechanistic connection between PHF19 reduction and H3K27me3 spreading remains speculative. If possible, include experimental validation such as PHF19 knockdown in virulent macrophages, to test whether this alone can recapitulate the H3K27me3 redistribution observed in attenuated cells.

We agree it's speculative and have removed all mention of PHF19 from the paper. There are 2 additional reasons for removing PHF19: first, we checked *phf19* expression by RT-qPCR in an independent attenuated Turkish vaccine line (Pendik: PMID: 1461683) and its virulent precursor and in contrast to Ode *phf19* expression was up regulated in attenuated Pendik (see below); second, in our reply to Reviewer #2 we present our bioinformatic analysis of publicly available raw data RNA-seq reads from two cows freshly infected with sporozoites of the same strain (ENA accession code: PRJEB65111) and in Cow69 *phf19* is down regulated, but not in cow71 (see Table on page 8). The expression profile of *phf19* therefore appears to vary between attenuated vaccine lines and even between different freshly infected cows.

Pendik

Legend: Expression of bovine *mmp9* and *phf19* in Vir and Attn strains from Ankara Pendik line. In the paper, we showed that *phf19* was more expressed in Vir Ode compared to Attn Ode. However, in the Turkish Pendik lines, it is more expressed in Attn than in VIR.

The paper identifies several unanswered questions (e.g., the role of *Theileria* in modulating H3K27me3, the impact of PHF19 reduction) but does not propose a clear roadmap for future research. Include a dedicated "Future Directions" section outlining specific experiments and approaches to address these questions. Briefly address the potential clinical relevance of its findings. For instance, could targeting PRC2 or PHF19 be a viable strategy for controlling *Theileria* infections or related cancers?

Our clear road map for future research does not include further characterization of PRC2-mediated H3K27me3 silencing, since our *in mice* study has shown that targeting PRC2 with the EZH2/EZH1 inhibitor UNC1999 is not a viable clinically relevant control strategy. UNC1999 treatment had no effect on *Theileria*-induced tumour dissemination in Rag2 γ C mice and using UNC1999 to treat cattle is difficult to imagine given the quantities required. In the future we would rather concentrate on other repressive marks that could potentially play a role in the loss of *Theileria*-transformed macrophage virulence and one such repressive mark for which we have very preliminary data is H3K9me3.

While the study identifies changes in H3K27me3 distribution, it does not fully elucidate the underlying mechanisms.

It's clear that in attenuated macrophages H3K27me3 marks spread in the absence of gain-of-function mutations in *ezh2*, where such a mutation has recently been shown to lead to a similar spreading of H3K27me3 in B-cell lymphomas (PMID: 38658543). To complete our reply to the reviewer, we have now checked for mutations in *ezh1* in virulent and attenuated macrophages, as either EZH1 or EZH2 can generate the H3K27me3 mark and the corresponding genes could have acquired gain-of-function mutations in attenuated macrophages and been responsible for the aberrant spreading (PMCID: PMC2630502). No amino acid changing mutations were found in *ezh1* or *ezh2* in either virulent or attenuated macrophages.

For example, what drives the broadening of H3K27me3 in attenuated macrophages?

We have added to the requested section “A clear road map for future research/Future Directions” the possible mechanisms that could lead to the abnormal spreading e.g., lines 248-251 “Although host EZH2 protein levels appeared equivalent in attenuated macrophages (Fig. 2a), other mechanisms such as phosphorylation^{61,62} and oxidative stress^{43,44} may contribute to heightened methyltransferase activity and/or decreased demethylase activity, thus resulting in spreading of the H3K27me3 mark in attenuated macrophages.” Furthermore, on lines 210-217 we added “Additionally, *T. annulata* is known to secrete a number of proteins predicted to be highly intrinsically disordered proteins (HIDP) such as TashAT2⁴⁷, Ta9⁴⁸ and the NIDP family⁴⁹, and all have been located to the host cell nucleus. Interestingly, in the context of chromatin silencing, overexpression of TashAT2 in BoMacs repressed transcription of 518 genes and upregulated transcription of 319 genes out of 837 DEGs⁴⁷. Moreover, when Ta9 was overexpressed in BoMacs⁵⁰ the top-scoring transcription factor predicted to regulate Ta9-responsive DEGs was MTF2, a component of the PRC2 complex⁵¹. So, secreted parasite HIDPs might contribute to the formation of phase-separated compartments in the chromatin of host cells^{52,53}.”

The study mentions that H3K27me3 maintains the silencing of key tumor suppressor genes, but does not elaborate on the functional consequences of this silencing. Suggestion: Discuss more about how the silencing of these genes contributes to the transformed phenotype of *Theileria*-infected macrophages and whether this has implications for the parasite's survival or dissemination.

We have only compared virulent to attenuated macrophages and both harbour parasites and differ in the ability of infected macrophages to disseminate. Therefore, we can't comment on parasite survival. However, we have followed the suggestion to look for tumour suppressor genes and extended the COSMIC comparison to genes repressed by H3K27me3 in the attenuated macrophages (29 genes). This found only PDGFRB (platelet derived growth factor receptor beta) and ACKR3 (atypical chemokine receptor 3) and neither has been reported as having tumor suppressor function. We also checked COSMIC for the presence of genes commonly repressed by H3K27me3 in both the virulent and attenuated macrophages (21 genes) and only PRF1 (perforin 1) was found. Concerning implications on dissemination we point out that the major established player *mmp9* is not marked by H3K27me3 and that regulated expression of *mmp9* appears dependent on changes in AP-1-driven transcription that we've shown is influenced by TGF- β -mediated fluxes in cAMP (PMID: 36847534; PMID: 25690101; PMID: 27835919).

Previously, we have demonstrated that GZMA and RASGRP1 could have novel tumor suppressor functions in attenuated macrophages, and, for GZMA, we have proposed that it acts via cleavage of APEX1. Cleavage of APEX1 would stop it from reducing redox sensitive cysteine residues of target transcription factors, so diminishing their DNA-binding and transcriptional activity (PMID: 32830401). However, we observed that expression of both *gzma* and *rasgrp1* was repressed by TGF- β virulent macrophages and consequently, expression increased in attenuated macrophages that have low TGF- β levels (PMID: 32830401). Expression of *apex1* is stimulated by TGF- β , whereas expression of *gzma*

and *rasgrp1* is repressed in virulent macrophages and this silencing does not appear to be sensitive to UNC1999 treatment (Table S2).

Reviewer #2 (Remarks to the Author):

In this manuscript, Sakura et al investigates the dynamics of the H3K27me3 chromatin mark during long-term passage of bovine macrophages infected with *T. annulata*. Through immunofluorescence staining (Fig. 1), Western blot (Fig. 2A) and ChIP-Seq (Fig. 2C-D), they observe that attenuated macrophages display an increase and altered pattern of H3K27me3 deposition vs virulent macrophages. Whereas in the latter, H3K27me3 is enriched at transcription start sites (TSS), in the former H3K27me3 was most enriched in intergenic regions. However, at a global level by RNA-Seq (Fig. 3), there were no statistical differences in expression of genes in virulent vs attenuated macrophages when sub-setted on different patterns of H3K27me3 enrichment. Although H3K27me3 is enriched in attenuated macrophages, pharmacological inhibition of PRC2 minimally de-represses gene expression in attenuated macrophages (in fact, PRC2 inhibition results in greater transcriptional de-repression in virulent vs attenuated macrophages) (Fig. 5) and does not rescue the invasive phenotype of attenuated macrophages in in vitro and in vivo assays (Fig. 4). The authors point to recently identified functional tumour suppressors (GZMA, FST) as PRC2-repressed genes (Fig. 5).

The authors identify interesting chromatin dynamics during the transition from virulent to attenuated phenotype of infected macrophages, yet whether these changes are functionally important for macrophage attenuation is unclear. Nevertheless, the manuscript could be strengthened in a few key areas:

1) Inclusion of uninfected control. Particularly for analysis of H3K27me3 levels and distribution by ChIP-Seq (Fig. 2), it would be informative to understand how H3K27me looks in uninfected cells. This would provide greater context for the H3K27m3 changes studied here between virulent and infected macrophages by establishing a baseline.

Uninfected primary macrophages do not proliferate unless stimulated and therefore would be difficult to use as a negative control for analysis of H3K27me3 levels and distribution by ChIP-Seq. In fact, having a good negative uninfected control is a problem when studying *Theileria*-induced macrophage transformation, as it is more generally in cancer research, where the primary tumour progenitor is often not available. For *Theileria*-transformed macrophages BoMacs are occasionally taken as a negative control (PMID: 39034325), but BoMacs have been immortalized by SV40 virus (PubMed:7676607). Thus, if one took BoMacs as a negative control one would be comparing the distribution of H3K27me3 marks between SV40-immortalized macrophages to the distribution in *Theileria*-transformed macrophages, and this is far from ideal.

Occasionally in *Theileria* research uninfected, but immortalized BL3 or BL20 cells are used as negative controls especially when testing parasite-specificity of drug doses (PMID: 36380082; PMID: 35882887). So, we performed a western blot comparing H3K27me3 levels between BL3 and *T. annulata*-infected BL3 (TBL3, see below) and this indicates that infection has led to down-regulation in H3K27me3 levels. However, as our focus was on determining if

there was a role for PRC2-mediated silencing of virulence in attenuated Ode macrophages we reasoned that the best control for attenuated Ode macrophages was to compare the distribution of H3K27me3 marks with that of virulent Ode macrophages, as they are isogenic and harbour the same *T. annulata* parasite strain. That said, and to satisfy the request of Reviewer #2 for a negative parasite-free macrophage control, we generated buparvaquone parasite cured macrophages. To this end, virulent *T. annulata*-transformed macrophages were treated with buparvaquone (BPQ), following the protocol described in (PMID: 24626571), to selectively eliminate the parasite without affecting host cell viability. Total RNA was extracted, and the expression levels of bovine *ezh1* and *ezh2*, as well as parasite p104 (used as a control for parasite clearance) were assessed by RT-qPCR. The results confirm that *Theileria* infection leads to the downregulation of both *ezh1* and *ezh2* expression at the mRNA and protein levels. This may enable parasite infection to rewire host cell gene expression (PMID: 22533473) and the increase in H3K4me3 signal (at the exposure used) is consistent with rewiring of the host transcriptome.

While analysing our own RNA-seq datasets (GEO ID: GSE135377; ENA accession code: PRJEB77601) we noticed a publicly available dataset (ENA accession code: PRJEB65111) derived from macrophages infected *in vitro* by *T. annulata* sporozoites. Since this data was not generated by us and is not yet been published, we present uniquely our bioinformatic analysis of PRC2 complex genes only as part of our rebuttal, as it confirms and extends our observations presented above that infection down regulates EZH2 and H3K27me3 marks. The two Tables show our analysis of the raw data reads for infected macrophages isolated from two cows (69 and 71). Highlighted in yellow are PRC2 complex genes that display significantly downregulated expression 3-weeks post-infection. *Mmp9* expression data is presented as a gene known to be induced by infection.

Cow #69

Gene Name	baseMean	log2FoldChange	padj
EZH1	797.828104238558	0.590768336969317	0.046365681163546
EZH2	1548.84089300272	-2.56538049056583	3.83949124463569e-21
EED	1147.19894012498	-0.645949499421851	0.0558820115361717
SUZ12	1222.80191828441	-2.78783828933251	3.39832636752985e-09
RBBP4	4113.49061829846	-1.20647052914768	0.000104701496829069
RBBP7	3126.62702710676	-1.94560325720648	5.88766104383248e-17
JARID2	1031.43424259646	0.756617088110395	0.0422147959911086
AEBP2	443.15801590862	-0.756448390207245	0.108544198347958
PHF19	387.702645209902	-1.77429537257286	4.35990962417274e-05
PHF1	1030.01053827626	1.63893430835422	9.43968203773253e-05
MTF2	227.216951476283	-1.49602106282904	1.23110707282517e-05
MMP9	110513.744800325	3.10645541722569	2.59684678610149e-12

Cow #71

Gene Name	baseMean	log2FoldChange	padj
EZH1	591.248618961051	0.277075047012494	0.510847543592625
EZH2	1575.09858827173	-2.35811001611461	4.85278990471848e-06
EED	796.614482755741	-0.565115141114366	0.142282111552145
SUZ12	932.936013423297	-1.97104781305849	2.19985563989724e-11
RBBP4	4085.40946098188	-0.92894979518002	0.00145859391768496
RBBP7	2770.15302959581	-1.96471727733042	9.84911342875392e-11
JARID2	849.475824368755	0.689767208299531	0.0425878833219728
AEBP2	443.611890911277	-0.739571626747821	0.0212897427661898
PHF19	286.875104207866	-1.09141069227612	0.193540477197893
PHF1	978.230832741669	1.66907563968683	2.24189814522978e-11
MTF2	218.377782580877	-1.29173137188036	4.97054658698151e-07
MMP9	279752.229338299	2.83945676993825	1.36806489454371e-11

Theileria infection downregulates core components of the PRC2 complex.

2) Quantification of immunofluorescent images. The increase in H3K27me3 levels in bulk attenuated macrophages is quite clear (Fig. 2). However, the immunofluorescent images shown in Fig. 1B present a unique opportunity to extend the analysis. At a single-cell level, how do H3K27me3 levels look in attenuated vs virulent macrophages? Is this mark uniformly increased in all attenuated macrophages or certain subsets? And is there any correlation with H3K27me3 levels and parasite burden (p104 staining)?

Single-cell immunofluorescence was performed to estimate the level of the H3K27me3 mark between virulent and attenuated Ode macrophages (see below).

Legend: Single-cell immunofluorescence quantification of the H3K27me3 mark between virulent (Vir) and attenuated (Att) Ode macrophages. The results support our claim in the paper, showing increased H3K27me3 levels in Att Ode. Red horizontal bars show the median.

To complement the IFA study, we queried the only available scRNA-seq data for *T. annulata*-transformed macrophages using paraCell: a novel software tool for the interactive analysis of dual host-parasite single-cell RNA-seq data (PMID: 39988320). This study found that >95% of macrophages were infected and paraCell identified different cell clusters based on co-expression of marker genes.

Clusters 0, 3, 8, and 11 were considered cycling macrophages; clusters 9 and 10 as tissue repair M2-like macrophages; clusters 2, 4, 5, and 12 as macrophages actively responding to the infection; clusters 6 and 7 as non-activated macrophages. Clusters 4 and 6 contained a higher percentage of parasites. Shown below are two violin plots, one for EZH1 and p104 (TA08425), and one for EZH2 and p104 (TA08425).

Based on this published scRNA-seq analysis, it appears that the degree of *ezh2* expression (reflective of H3K27me3 marks) does not parallel parasite load (TA08425 = p104 expression). We provided further evidence to support this through single-cell analysis of p104 and H3K27me3 signals in confocal images, and their correlation in attenuated Ode macrophages (see below).

Legend: Parasite load (intensity of p104 signal) does not correlate with H3K27me3 levels. This was tested in attenuated (Att) Ode macrophages, which displayed a more heterogeneous H3K27me3 signal. These results are consistent with scRNA-seq data we extracted from an independent *T. annulata*-transformed macrophage line (PMID: 39988320).

3) Further integration of ChIP-Seq and RNA-Seq data. In Fig. 3, the authors note a small number of genes which have lower expression in attenuated macrophages coinciding with the increased H3K27me3 in these cells. Upon further inspection of SKAP2, which is downregulated in expression in attenuated macrophages, this gene locus not only gains H3K27me3 but also loses H3K4me3 (Fig. 3C). H3K4me3 levels don't appear to be globally different in virulent vs attenuated macrophages (Fig. 3B) but are DEGs characterized by gain of H3K27me3 AND loss of H3K4me3?

As mentioned in the text of the submitted manuscript, the tracks of H3K4me3 around the TSS showed no global difference between virulent and attenuated macrophages in all three clusters (Fig. 3a). Thus, we believe that the loss of H3K4me3 is less likely to be involved in transcriptional silencing of the corresponding gene. Other epigenetic mechanisms of silencing, such as those mediated by histone marks or DNA methylation, may be important.

Furthermore, is the RNA-seq data used here generated by the authors or from the literature? This should be made more clear.

Above in reply to Reviewer #1 we pointed out that the RNA-seq data on virulent and attenuated Ode macrophages was generated by us and cited in the submission as reference 35.

In a similar fashion, are there any defining chromatin features of genes subject to PRC2-mediated repression in Fig. 5? Integration of these data with ChIP-seq could yield some insights.

The 50 DEGs identified in attenuated macrophages by RNA-seq upon PRC2 inhibitor treatment were integrated with ChIP-seq. Eight out of 50 (CD69, IRF1, TMEM108, ENC1, CHN1, EHD3, GZMA, LY6E) belong to cluster 2 in Fig. 3a, and other genes are categorized into clusters 1 and 3. However, from these analyses we couldn't define any clear chromatin

features. Heatmaps and signals of both H3K27me3 and H3K4me3 marks on the DEGs in each group were visualized in Fig. S3 and we added the text to the Discussion (lines 222-226).

4) Readout of in vivo assay. In Fig 4, the authors look at parasite burden in heart and lung tissues as their readout after PRC2 inhibition in attenuated macrophages. Did the authors look at engraftment or dissemination of the macrophages themselves?

We did not look at engraftment, but did look at dissemination of bovine macrophages by detecting the presence of the parasite, as >95% of transformed bovine macrophages harbour *T. annulata*.

5) Connection with TGF- β . The authors cite that TGF- β 2 signaling is a known signal that can restore the dissemination capacity of attenuated macrophages, suggesting that epigenetic changes may contribute to the attenuated phenotype. Can the authors expand on why they believe TGF β may be working via the epigenome? Further yet, it could be interesting for the authors to look at H3K4me3 and H3K27me3 in attenuated macrophages treated with TGF β . Regardless of the result, it could shed some light on the mechanisms underlying TGF β in establishing the attenuated macrophage phenotype and the importance of epigenetics in this TGF β -mediated process.

We don't believe that the impact of TGF- β on increasing dissemination of attenuated macrophages is exclusively mediated by epigenetics, because adding exogenous TGF- β to attenuated macrophages increases AP-1-driven transcription of *mmp9*. We note that in attenuated macrophages down-regulated *mmp9* is not marked by H3K27me3. Although *mmp9* is not marked by H3K27me3 in the ChIP-seq data, it's commonly upregulated in both virulent and attenuated Ode macrophages upon PRC2 inhibition (Table S2, Sheet 1). We mentioned that TGF- β signalling might stimulate epigenetic changes due to its ability to induce DNA methylation of certain target genes (PMID: 30618784). This suggests that regulation of expression of *mmp9* is complex, or even that inhibitor UNC1999 may be exerting secondary effects via other methyltransferases, such as SMYD3, which has been implicated in regulation of *mmp9* expression in *Theileria*-transformed leukocytes (PMID: 22194464).

Minor comments:

1) Fig. 5C is a very untraditional way to display pathway analysis. Please either perform gene set enrichment (GSEA) or show p-values for top pathways.

2) In some RNA-Seq legends and ChIP-Seq tracks, the axes labels are quite small and could be increased in size to enhance readability.

We've prepared an additional Excel supplementary table (Table S4) that includes all p-values of each pathway. The label sizes of RNA-seq and ChIP-seq have been increased.

Reviewer #3 (Remarks to the Author):

The authors have attempted to look at the molecular events that underpin the attenuation of *Theileria annulata* infected cells. Specifically, they've looked at H3K27me3 and H3K4me3 modification profiles of pathogenic vs attenuated cells. Generally, the interesting

observation is the re-distribution of H3K27me3 from tight peaks at TSS in pathogenic to spread around TSS in attenuated cells. Although the biological significance of this observation, in the context of the pathogenic vs attenuated phenotype, has not been demonstrated considering that they've shown it not due to significant changes in gene expression. The work appears straightforward, except it would have been nice to either show, or provide reference, phenotypic evidence that the cells are indeed pathogenic or attenuated. While there's RNA-seq analysis on PRC2-inhibitor treated cells, it would have been good to show expression levels of genes like SKAP2 that show the changes in H3K17me3 distribution in attenuated cells. There is a reference to a supplementary table 2 with expression levels, but I didn't see it. There discussion seem to introduce new results that should be included in the results section rather than discussion e.g., line 211-214. Line 155 has (ref), which I assume was a placeholder for the author to include a reference. As there's no statistical test, the western blot results on Fig2A, should be described as numerical higher on line 114.

Since the number of DEG is limited, most gene-expression changes were not significant, as noted in our reply to Reviewer #2 above. The read counts of SKAP2 in attenuated macrophages treated with either UNC2400 or UNC1999 were too low (less than 10) to quantify.